# A Sensitive Band to Optimize Winter Wheat Crop Residue Cover Estimation by Eliminating Moisture Effect

**Yamei Wang** [1,2,3,4], **Shuhe Zhao** [1,2,3,5,*], **Wenting Cai** [1,3,4], **Joon Heo** [5,*] and **Fanchen Peng** [1,3,4]

1   School of Geography and Ocean Science, Nanjing University, Nanjing 210023, China;
    mg1827104@smail.nju.edu.cn (Y.W.); mg1727099@smail.nju.edu.cn (W.C.);
    mg1827068@smail.nju.edu.cn (F.P.)
2   Jiangsu Center for Collaborative Innovation in Geographical Information Resource Development and
    Application, Nanjing 210023, China
3   Collaborative Innovation Center of South China Sea Studies, Nanjing University, Nanjing 210023, China
4   Jiangsu Provincial Key Laboratory of Geographic Information Science and Technology, Nanjing University,
    Nanjing 210023, China
5   Department of Civil and Environmental Engineering, Yonsei University, Seoul 100-744, Korea
*   Correspondence: zhaosh@nju.edu.cn (S.Z.); jheo@yonsei.ac.kr (J.H.)

**Abstract:** Crop residues can retain soil moisture and increase soil organic matter. Crop residue cover is also a hot issue in agricultural remote sensing. Crop residue cover can be estimated linearly with cellulose absorption index (CAI), while moisture of crop residues and soil would reduce the accuracy of crop residue cover estimation. Crop residue and soil were used as materials to carry out the laboratory experiment to reveal the impact of moisture on crop residue cover estimation and eliminate said impact. This paper discovered a sensitive band, $R_{2005}$, which can invert water content of materials to eliminate moisture effect and improve estimation accuracy of crop residue cover. In terms of inverting water content, compared with two ratio water indices proposed in 2016 ($R_{1.6}/R_{1.5}$, $R_{1.6}/R_{2.0}$), using $R_{2005}$ can increase $R^2$ from 0.828 to 0.935 and decline root-mean-square error (RMSE) from 0.12 to 0.07. At the point of results validation, $R^2$ is 0.958 and RMSE is 0.06, indicating $R_{2005}$ has a high accuracy. Another advantage of $R_{2005}$ is that it is more suitable to promote to actual production because of simple and efficient band calculation.

**Keywords:** winter wheat; crop residue cover; cellulose absorption index (CAI); $R_{2005}$

## 1. Introduction

Crop residues refer to the crop stubble in the cultivated land after harvest. Keeping crop residues in the farmland is one of the best agricultural management measures [1]. Besides reducing the economic cost of removing crop residues, this method also has important ecological implications. A large number of studies have shown that soil organic carbon content in farmland is the basis and guarantee for a high yield of crops [2–5]. Crop residues remaining in cultivated land play an important role in promoting soil carbon sequestration and improving soil fertility. It has been reported that the soil carbon pool will increase by 8%~35.7% in organic carbon content after maintaining crop residues in the crop land [6,7]. In addition, crop residue can also play a "protective film" role. Crop residues covering the surface can effectively reduce the evapotranspiration of surface water and prevent soil erosion [8,9]. There are other studies, which have shown that the more crop residues left on soil surface, the stronger the ability to regulate soil PH values and inhibit crop diseases [10,11]. In general, crop residue cover on farmland is critical to agricultural development.

Accurate estimation of crop residue cover in farmland has always been an agricultural focus. Remote sensing techniques have been widely used to estimate crop residue cover, and many crop residue indices have been constructed. Zheng et al. [12] used multi-temporal Landsat TM and ETM+ imageries to develop minimum normalized difference tillage index (NDTI) to map crop residue cover commendably. Sullivan et al. [13] found that crop residue cover could be well reflected by differentials in blue and near-infrared bands, and they proposed crop residue cover Index 1 (CRC1). Qi et al. [14] proposed the normalized differential senescent vegetation index (NDSVI) that is based on the normalized differential of TM3 and TM5. Gelder et al. [15] introduced the normalized difference residue index (NDRI) based on TM3 and TM7, the effect of which is better than NDTI with the presence of vegetation. At the same time, they [15] proposed a statistical correction method of green vegetation, based on normalized differential vegetation index (NDVI), to improve the accuracy of crop residue cover estimation. Based on the cellulose absorption valley at 2100 nm, Daughtry et al. [16] proposed cellulose absorption index (CAI), and laboratory data confirmed that CAI could separate soil and crop residues well. Then, lignin cellulose absorption index (LCA) and shortwave infrared normalized difference residue index (SINDRI) based on NASA Terra Advanced Spaceborne Thermal Emission and Reflection Radiometer (ASTER) data were introduced [17,18].

The above crop residue indices can improve the estimation accuracy of crop residue cover while remaining very sensitive to soil background. In order to eliminate the influence of soil background, Biard et al. [19] used TM4 and TM5 bands, added soil line, and proposed soil adjusted corn residue index (SACRI) to reflect the information of crop residue information under low coverage. Bannari et al. [20] used two infrared channels to modify SACRI and proposed modified soil adjusted corn residue index (MSACRI). This index does a better job of weakening the soil background effect. Biard and Baret [21] introduced a linear mixture model, crop residue index multiband (CRIM), which is based on the soil and crop residue line. The model can combine any bands and apply to high or low resolution images for uniform land surfaces.

Moisture factors of crop residues and soil will affect the accuracy of crop residue cover estimation. Since water affects the spectral characteristics of crop residue and soil, and thus the estimation of crop residue cover, it is necessary to eliminate the effects of moisture. In the area of this scientific progress, some studies have used spectral water indices and related crop residue indices. Quemada et al. [22] proposed to use spectral water indices to simulate crop residue and soil moisture and explored the influence of moisture on crop residue cover estimation. The results showed that, among the three narrow-band ratio water indices ($R_{1.6}/R_{1.5}$, $R_{1.6}/R_{2.0}$, $R_{2.2}/R_{2.0}$) and two wide-band ratio water indices (SWIR3/SWIR6, OLI6/OLI7), $R_{1.6}/R_{1.5}$ and $R_{1.6}/R_{2.0}$ have better simulation effect on water. The specific information of above water indices is shown in Table 1. Quemada et al. [23] used Worldview-3 images to simulate crop residue cover in the field. Partial irrigation of farmland is needed to complete the comparative experiment. First, the ground objects were classified according to the digital photos taken in the field to obtain the coverage of crop residue. NDTI and SINDRI of dry farmland were calculated by bands calculation of satellite images. The relationships between crop residue cover and NDTI, SINDRI were obtained by linear regression analysis of crop residue cover and indices under dry conditions. For the farmland area with irrigation, the moisture correction model including the water index was used to correct the spectral values of the images. After calculating the corrected NDTI and SINDRI, the researchers substituted the corrected NDTI and SINDRI into the model, which is not affected by moisture, to eliminate the effect caused by moisture.

**Table 1.** Selected bands of different water indices.

| Water Indices | The Selected Bands |
|---|---|
| $R_{1.6}/R_{1.5}$ | $R_{1.5}$ is 10-nm band centered at 1500 nm, $R_{1.6}$ is 10-nm band centered at 1600 nm |
| $R_{1.6}/R_{2.0}$ | $R_{1.6}$ is 10-nm band centered at 1600 nm, $R_{2.0}$ is 10-nm band centered at 2030 nm |
| $R_{2.2}/R_{2.0}$ | $R_{2.2}$ is 10-nm band centered at 2200 nm, $R_{2.0}$ is 10-nm band centered at 2030 nm |
| SWIR3/SWIR6 | SWIR3: 1640–1680 nm, SWIR6: 2185–2225 nm for WorldView-3 |
| OLI6/OLI7 | OLI6: 1570–1650 nm, OLI7: 2110–2290 nm for Landsat8 |

At present, the existing water indices cannot accurately indicate the water content of crop residue and soil, resulting in a large error in study results. Therefore, a sensitive band was found in this paper, which can simulate the true moisture condition with higher accuracy. Our goal was to invert water content of crop residue and soil precisely to further estimate crop residue cover precisely.

## 2. Materials and Methods

### 2.1. Laboratory Experiment

#### 2.1.1. Crop Residue

To study and eliminate impact of moisture on crop residue cover estimation, winter wheat residues and soil collected in Liaocheng City of Shandong province, China, were used as materials to carry out the laboratory experiment. Liaocheng City, which belongs to Huang-Huai-Hai Plain, is the main winter wheat producing area in China.

First, five trays of crop residues were dried until they had a constant mass, which is the dry weight of the crop residues. Then, each crop residue sample was soaked in water overnight. After soaking, the crop residues had reached saturation and could not absorb any more water. The excess water in samples was drained and samples were weighed out to calculate water content. This paper selected relative water content (RWC) as the index to measure the water content [24]. RWC was calculated by Formula (1),

$$\text{RWC} = \frac{m_{fresh} - m_{dry}}{m_{max} - m_{dry}} \tag{1}$$

where $m_{fresh}$ is the sample mass at the time of weighing, $m_{dry}$ is the dry weight of sample, and $m_{max}$ is the saturation weight of sample.

Crop residues reflectance spectra were acquired by ASD spectroradiometer over range of 350–2500 nm. The sampling interval was 1 nm. When the spectral data were obtained, each crop residue sample was placed in a black plastic tray with a depth of 2 cm and a diameter of 10 cm. To avoid external influence, the experiment was conducted in a dark room, and the light sources were quartz-halogen lamps that could provide 1000 W parallel light. The light beam was over the samples at a 45° zenith angle and at the height of 50 cm from samples surface. The zenith angle was set 0°, and vertical distance from samples surface was 30 cm, A fiber-optic probe with a view angle of 5° was arranged to acquire samples' reflectance spectra. The beam angle and view angle were set to minimize shadows and highlight the spectral characteristics of samples. Four spectra of five scans each were required by rotating the sample tray 90° after each measurement to reduce specular reflection that might exist in a laboratory experiment. Before each spectra acquisition, a Spectralon reference panel was needed to be placed in the field of view, illuminated, and measured in the same manner as the samples.

To slow the rate of moisture loss, crop residue samples were placed at room temperature to be dried. The samples were weighed out and the reflectance spectra were acquired at 2 h intervals until the mass of samples was close to dry weight. Before the end of experiment, when the samples were dried to lose all water again, final weighing and reflectance collection were carried out.

### 2.1.2. Soil

In order to ensure the uniformity of soil particles and prevent the influence of particle size on spectral data, the soil samples should be fully ground before the experiment. In accordance with the crop residues, five trays of soil were prepared as experimental materials. The soil samples were dried to obtain dry weight, then soaked overnight to allow the soil and water to mix well, filtered for excess water, and weighed. When the ASD spectroradiometer was used to obtain spectral data, the soil samples were placed in pure black plastic trays with depth of 2 cm and diameter of 10 cm. The times of soil spectral measurements and weighing were consistent with crop residue. Measurement methods were strictly taken according to crop residue samples data acquisition standards. The flow chart of the laboratory experiment is shown in Figure 1.

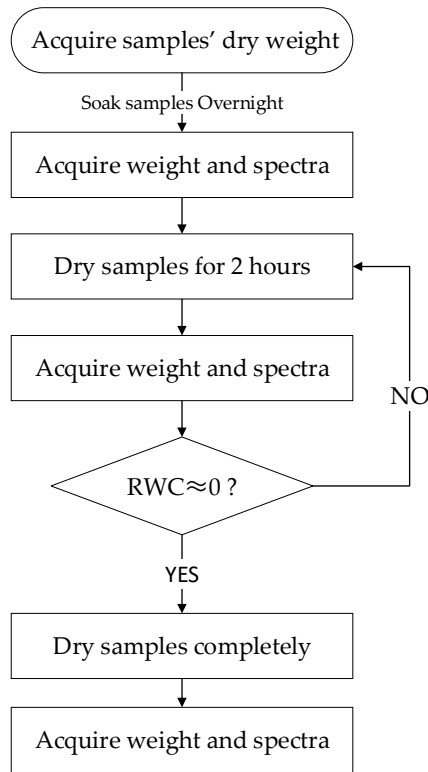

**Figure 1.** The flow chart of experiment.

### 2.2. Data Processing

### 2.2.1. Preprocessing

ViewSpecPro 6.0 software was used to process the spectral data. We visually judged and deleted the spectral curve with the largest difference of each sample datum and took the mean value of the remaining spectral curves. In order to eliminate the noise interference, the Savitzky–Golay filter is used to smooth the graph to get the final result.

### 2.2.2. Spectra and RWC Data Processing

In practice, the crop residues could not cover all the farmland, so the collected reflectance in the field was affected by both soil reflectance and crop residues reflectance. In this paper, the reflectance of actual crop residues in farmland was simulated by a linear combination of crop residues spectra and soil spectra. The applied function is shown in Formula (2) [22,25],

$$\rho_{(M,\lambda)} = \rho_{(S,\lambda)}(1 - f_R) + \rho_{(R,\lambda)}(f_R) \tag{2}$$

where $\rho_{(M,\lambda)}$ is the simulated reflectance of the actual crop residue in farmland in $\lambda$ band, which can be regarded as the true spectral value; $\rho_{(S,\lambda)}$ is the reflectance of soil in $\lambda$ band; $\rho_{(R,\lambda)}$ is the reflectance of crop residues in $\lambda$ band; $f_R$ is the assumed value of crop residue cover and can be regarded as true observation value of coverage. $f_R$ in this study was acquired randomly by an algorithm.

For RWC data processing, like spectral data, the method of linear combination of the RWC of crop residues and the RWC of soil obtained with weighing method was also adopted to simulate the overall RWC of crop residues in the field. The Formula is (3) [22],

$$RWC_m = RWC_S(1 - f_R) + RWC_R(f_R) \tag{3}$$

where $RWC_m$ is the overall water content of crop residues under simulated field condition, which can be regarded as the true value collected in field; $RWC_S$ is the RWC of soil; and $RWC_R$ is the RWC of crop residues. $f_R$ is the assumed value of crop residue cover and can be regarded as true observation value of coverage. In the case of linear combination of spectral reflectance and RWC of crop residues and soil, in order to fully simulate the field conditions, the time of moisture reception and air drying for crop residues and soil should be consistent. Therefore, only the crop residue and soil data measured at the same time should be combined. The $f_R$ of Formulas (2) and (3) should be consistent when the spectral data and RWC data acquired are combined linearly.

### 2.3. Data Analysis Method

### 2.3.1. Crop Residue Cover Estimation Method

CAI, constructed based on hyperspectral data, was selected to establish the correlation with crop residue cover, and the equation of CAI was expressed as Equation (4) [22],

$$CAI = 100(0.5(R_{2.0} + R_{2.2}) - R_{2.1}) \tag{4}$$

where $R_{2.0}$, $R_{2.1}$ and $R_{2.2}$ are the reflectance of 10 nm bands centered at 2030 nm, 2100 nm, and 2210 nm, respectively. The relationship between CAI and crop residue cover was determined by unitary regression analysis. The essential function of CAI is to reflect the absorption intensity of crop residues spectra at 2100 nm or to roughly calculate the depth of the absorption valley at 2100 nm, which is the absorption characteristic position caused by the large amount of cellulose in crop residues. Therefore, there was a strong correlation between CAI and crop residue cover. The reason why spectral reflectance at 2000 nm was not selected is that carbon dioxide in the atmosphere has a narrow absorption valley at 2010 nm. Application of spectral data at 2030 nm can minimize the influence of the atmosphere [18].

### 2.3.2. Improvements to Crop Residue Cover Estimation Model

It had been proved that CAI and crop residue cover were linearly correlated, and crop residue cover can be estimated by a linear model with CAI [18,22,25,26]. But RWC of crop residues and soil influenced the slope and intercept of the function of the model [22,26]. There were different slopes and intercepts under varied RWC. If we want to estimate crop residue cover accurately, slopes and intercepts under varied RWC must be acquired. The relationship between the RWC, the slope, and the intercept of the model were analyzed by regression methods.

### 2.3.3. Sensitivity Analysis of RWC and Reflectance

This research calculated the sensitivity of the $RWC_m$ and the $R_{(M,\lambda)}$ over the range of 350–2500 nm to determine the sensitive band. The reflectance of the sensitive band was affected by moisture and could invert moisture.

### 2.3.4. RWC Inversion by Water Indices and Sensitive Band

The RWC of the soil and crop residue samples were calculated by weighing method. The simulated field moisture was obtained by linear combination of the RWC of the soil and crop residue samples. However, these methods cannot be extended to the field, so we needed to use remote sensing means to achieve the RWC simulation. A study used ratio water indices to indicate RWC in 2016 [22]. This paper also detected a sensitive band to invert RWC. The water indices proposed in 2016 [22] were compared with the sensitive band to verify the RWC inversion effects of the sensitive band.

### *2.4. Validation*

In order to further verify the accuracy of the model, 70 sample points were selected for model validation in this paper. First, CAI can be calculated by spectral data, and the $RWC_m$ of the two kinds of samples was inverted by water indices and the sensitive band. Second, the slope and intercept of crop residue cover estimation model were calculated under different moisture conditions. Finally, the crop residue cover was simulated by the crop residue cover estimation model. The three estimation results, which improved by two water indices and the sensitive band, were compared with the true crop residue cover obtained randomly through the algorithm.

## 3. Results

### *3.1. Analysis of Crop Residues and Soil Spectra*

In total, we acquired spectral data eight times under different moisture conditions. The spectra of crop residues and soil samples are shown in Figures 2 and 3. The gaps are spectral reflectance at 1350–1430 nm, 1795–1970 nm and 2400–2500 nm. The spectral values in these three ranges would be affected by the moisture in the air that would produce noise, which would affect the spectral characteristics of the samples. Therefore, these parts of the spectral value were deleted.

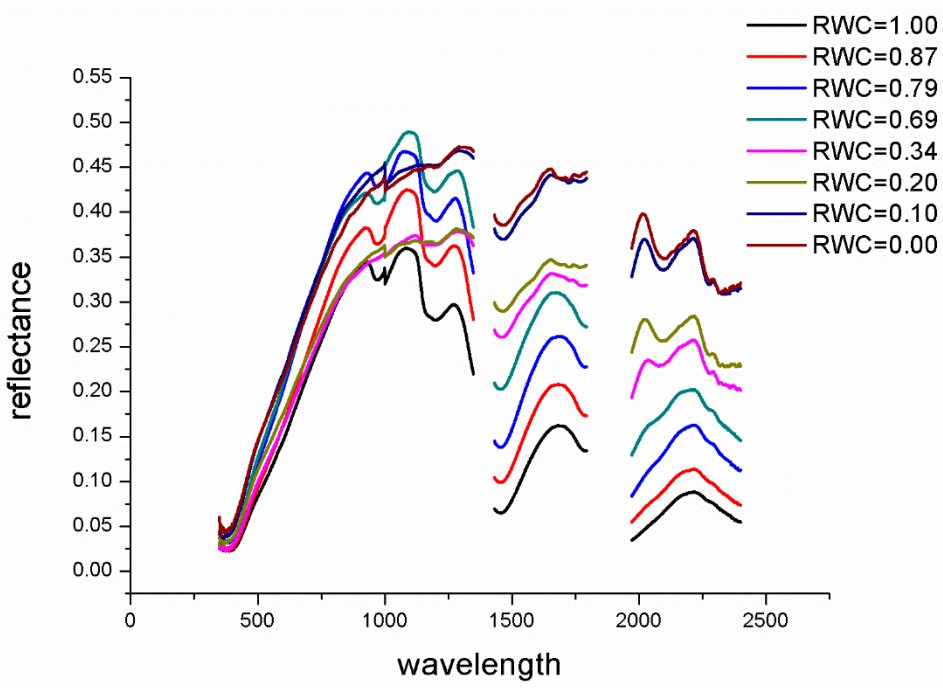

**Figure 2.** Reflectance of crop residue under various relative water content (RWC).

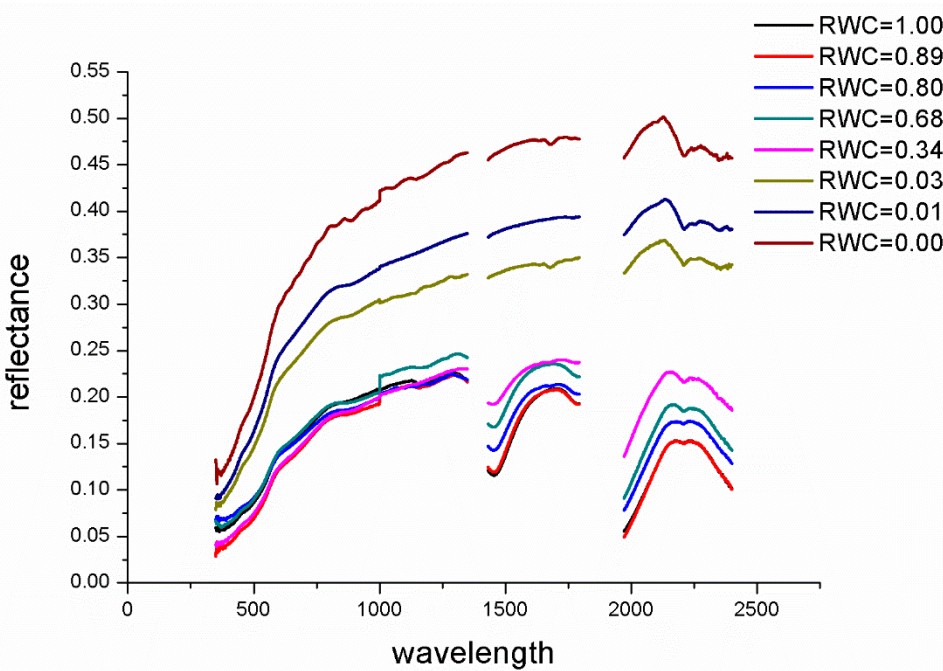

**Figure 3.** Reflectance of soil under various RWC.

As can be seen from Figure 2, the total spectral values of crop residue samples decreased with the increase of moisture. Moreover, as the wavelength increased, the spectral difference caused by moisture increased, which was because the absorption intensity of water in the infrared band is greater than that in the visible bands. When the moisture of crop residue increased, some of its spectral characteristics also disappeared, and the curves became more and more gentle. For example, the absorption intensity of cellulose at 2100 nm gradually weakened and disappeared when the overall spectral reflectance dropped. In the ranges of 1700–1795 nm and 2280–2400 nm, the spectral curves of crop residue with low moisture had zigzag features, while in the case of high moisture, the "zigzag" of the spectral curves disappeared, and smooth spectral features were presented. However, under any moisture condition,

the high reflection peaks at 1640 nm and 2210 nm always existed, and the higher the RWC, the more obvious the peak characteristics.

Figure 3 is the spectral characteristic curves of soil samples under different moisture conditions. Similar to crop residues, the higher the RWC, the lower the total spectral reflectance of the soil. Since the crop residue samples and soil samples were weighed and their spectra were collected at the same time, it can be seen that the rate of water evaporation of crop residue and soil were roughly the same when the moisture was high. While at the later stage of drying, the rate of soil water loss was significantly higher than that of crop residues, and a similar situation has occurred in another study [22]. The absorption valley at 2200 nm in the dry soil spectra was caused by the stretching vibration of hydroxyl (−OH) molecule of silicate water [27]. When the RWC increased, the absorption characteristic at 2200 nm became very weak. However, when the RWC increased, the strong absorption characteristics of the water at 1450 nm and 1960 nm will make the soil have a very obvious peak value in the range of short-wave infrared (1600~1780 nm), which can be used as the basis to roughly judge the soil water content [22]. In addition, excessive water also caused a significant single-peak phenomenon in the range of 2100–2330 nm of the soil spectra.

### 3.2. Crop Residue Cover Estimation Model and Moisture Effect

In this paper, the actual mixed reflectance of crop residues and soil under different coverages were not collected in the field. Instead, the individual spectral reflectance of crop residues and soil were acquired in the laboratory, and the actual spectral values in the field were simulated by linear combination. Under different moisture conditions, the relationships between spectral reflectance and CAI is shown in Figure 4.

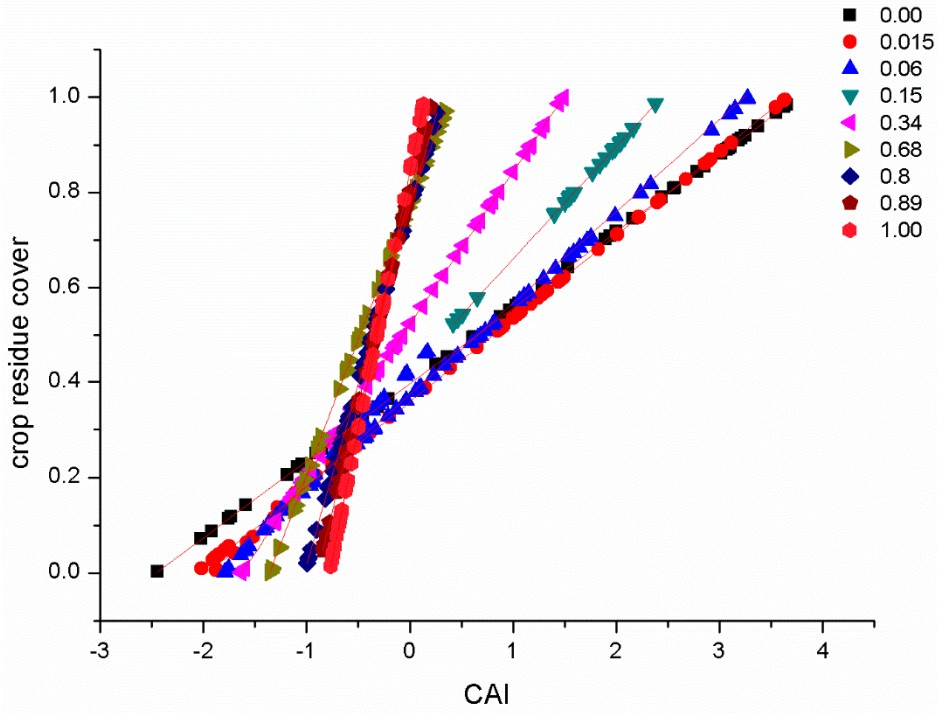

**Figure 4.** Crop residue cover as a function of cellulose absorption index (CAI) under various RWC.

In Figure 4, the abscissa is CAI, and the ordinate is the crop residue cover obtained randomly. The legend represented different moisture conditions, and the straight line was the unitary regression line of CAI and crop residue cover. It was clear that CAI and crop residue cover had a significant linear relationship with any moisture. Therefore, a linear model including CAI can be used to simulate crop residue cover. The slope and the intercept remained positive. While the RWC varies, the slope and intercept of the linear model are different. As the RWC increases, the slope and intercept increase, which

means that moisture would result in overestimation of crop residue cover. The influence of moisture on crop residue cover estimation can be determined and eliminated by finding the relationships between the RWC, slope, and intercept.

### 3.3. Inversion of the Slope and Intercept of Model

Obviously, there were different slopes and intercepts under varied RWC. Regression analyses were conducted on the RWC and the slope, the intercept of the model. The results are shown in Figures 5 and 6.

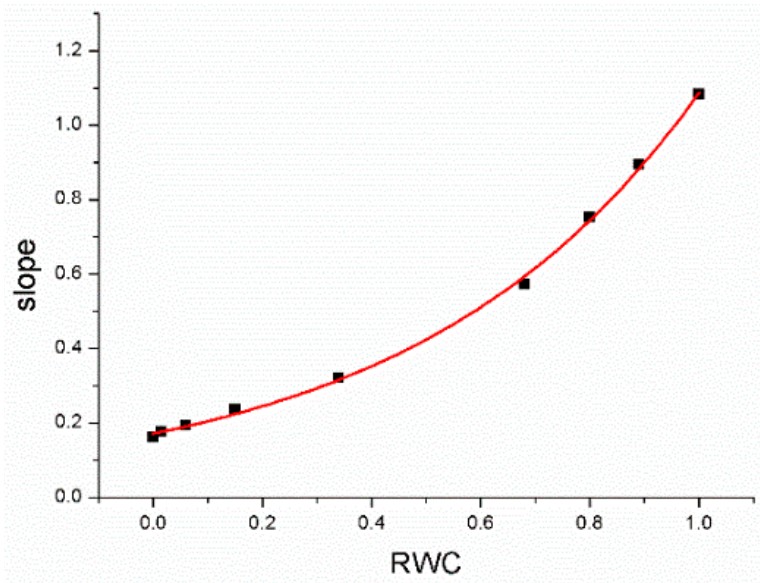

**Figure 5.** Slope as a function of RWC.

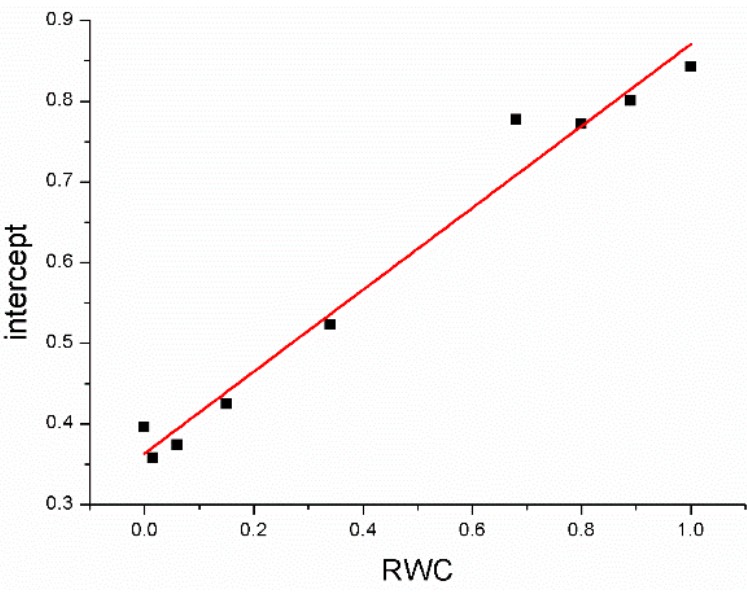

**Figure 6.** Intercept as a function of RWC.

There were definite functional relationships between the RWC, the slope, and the intercept of the model. RWC and slope conformed to a single exponential decay relationship. RWC was linearly dependent on the intercept. The specific functional relationships are shown in Table 2.

**Table 2.** Slope and intercept as the functions of RWC.

| Parameter | Equation | Adj.R$^2$ | RMSE |
|-----------|----------|-----------|------|
| slope | $\text{slope} = 0.15471 * \exp\left(\frac{\text{RWC}}{0.51696}\right) + 0.01725$ | 0.99868 | 0.01268 |
| Intercept | $\text{intercept} = 0.50728 * \text{RWC} + 0.36342$ | 0.9743 | 0.03336 |

### 3.4. Sensitivity Analysis of RWC and Reflectance

In this paper, correlation analysis was conducted between the RWC and spectral reflectance to determine the sensitive band, and the results are shown in Figure 7.

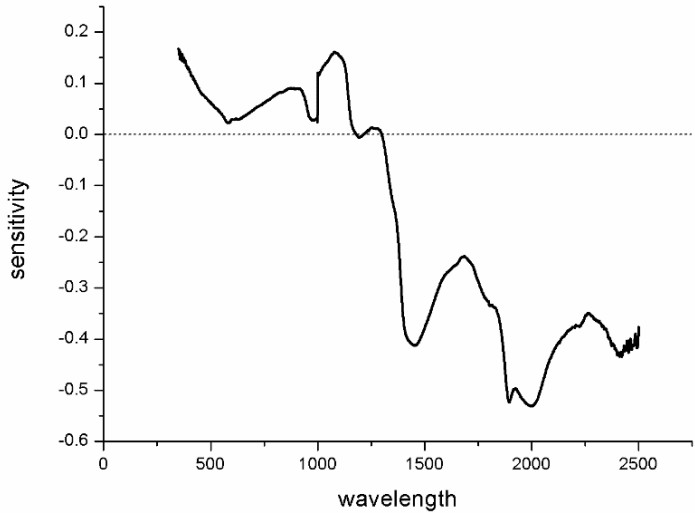

**Figure 7.** Sensitivity between RWC and reflectance.

There was no obvious regular correlation between RWC and spectral values in the whole range. Spectral values of the band range from 350 nm to 1180 nm are positively correlated with RWC, but correlation coefficients, which were less than 0.2, were generally small. The correlation coefficients at 350 nm and 1070 nm were relatively high, and were almost irrelevant around 580 nm. Therefore, the wavelength ranges that were less than 1180 nm had little significance for the inversion of RWC. There were negative correlations between sample reflectance and RWC from 1300 nm to 2500 nm. There were strong correlations at 1460 nm, 1900 nm, 2005 nm, and 2420 nm, and the absolute values of the correlation coefficients were significantly higher than those of the adjacent bands. The spectral reflectance of 2005 nm band is the most correlated, close to 0.6. Therefore, this single-band spectral reflectance was used as the basis for the inversion of RWC. The selected sensitive band was centered at 2005 nm with a 10 nm width ($R_{2005}$). In order to verify the accuracy of this sensitive band, two proposed water indices were chosen in this paper for reference and comparison. The specific information of each water index and the sensitive band is shown in Table 3.

**Table 3.** The information of water indices and the sensitive band.

| Water Indices/The Sensitive Band | The Selected Bands |
|----------------------------------|--------------------|
| $R_{1.6}/R_{1.5}$ | $R_{1.5}$ is 10-nm band centered at 1500 nm<br>$R_{1.6}$ is 10-nm band centered at 1600 nm |
| $R_{1.6}/R_{2.0}$ | $R_{1.6}$ is 10-nm band centered at 1600 nm<br>$R_{2.0}$ is 10-nm band centered at 2030 nm |
| $R_{2005}$ | $R_{2005}$ is 10-nm band centered at 2005 nm |

### 3.5. Inversion of RWC of Soil and Crop Residue

A study proposed three ratio water indices to invert RWC [22]. Since the inversion effect of $R_{2.2}/R_{2.0}$ was not ideal in the original text, only $R_{1.6}/R_{1.5}$ and $R_{1.6}/R_{2.0}$ were selected for comparison with the $R_{2005}$ in this paper. The results of fitting the individual RWC of soil and crop residue samples with $R_{2005}$, $R_{1.6}/R_{1.5}$ and $R_{1.6}/R_{2.0}$ are shown in Figure 8.

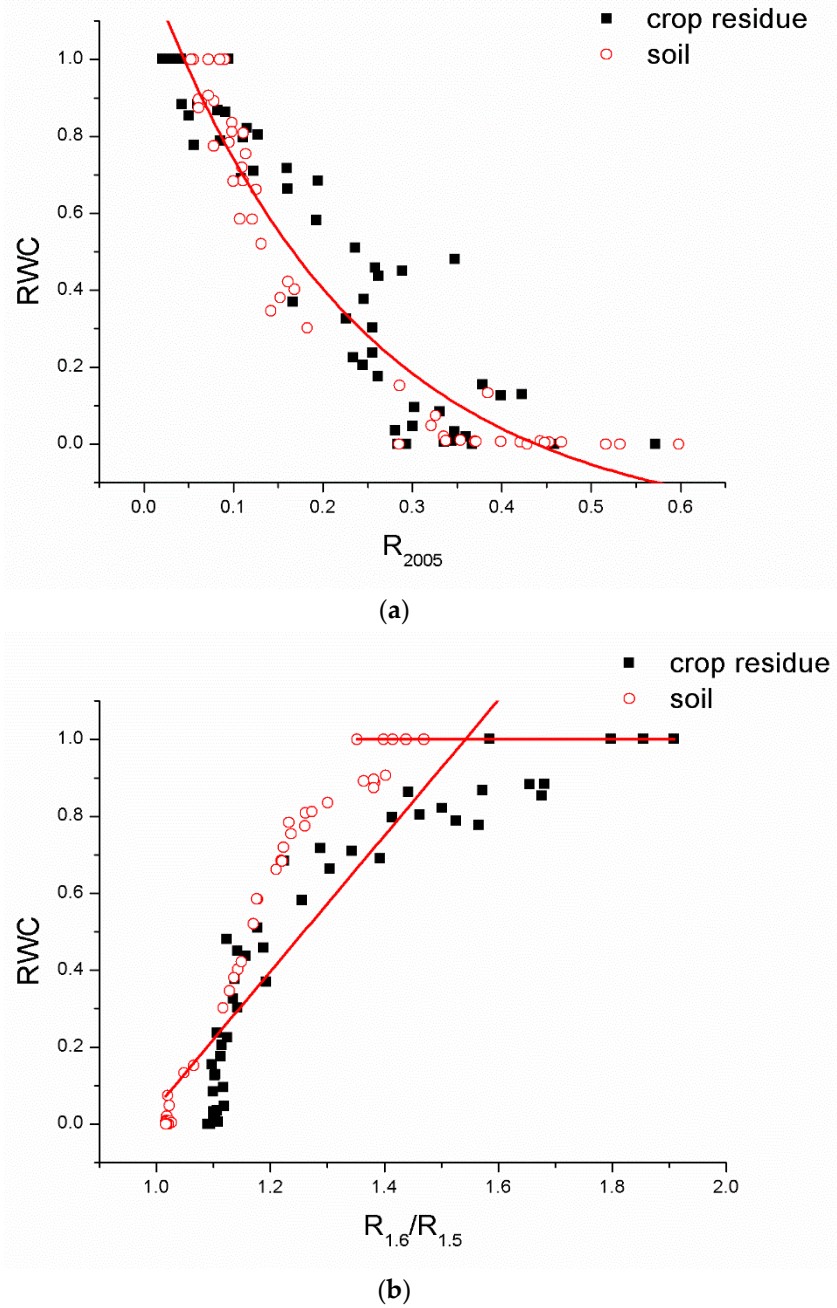

(**a**)

(**b**)

**Figure 8.** *Cont.*

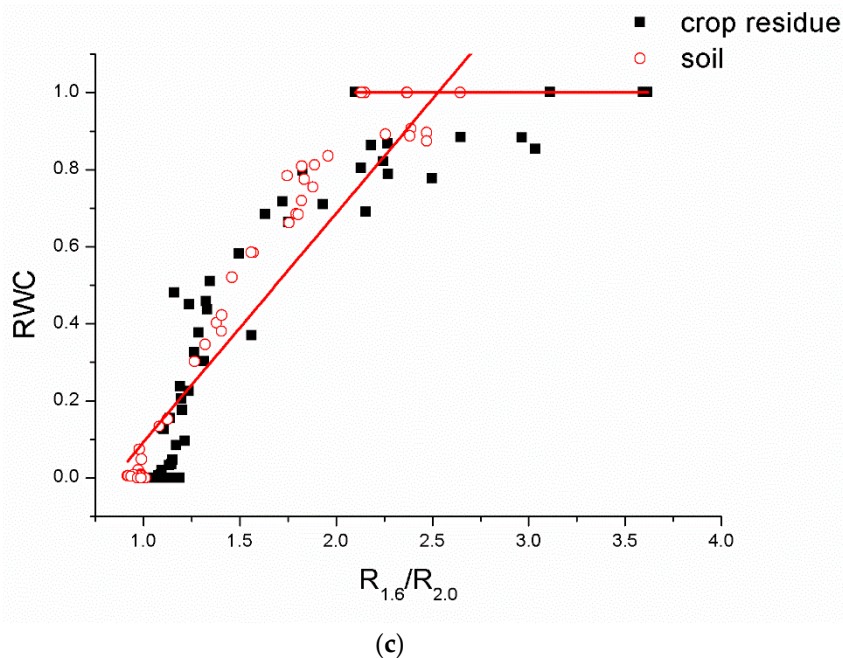

(**c**)

**Figure 8.** RWC of crop residues and soil as a function of the sensitive band and two water indices: (**a**) $R_{2005}$ (**b**) $R_{1.6}/R_{1.5}$ (**c**) $R_{1.6}/R_{2.0}$.

In the case of using $R_{2005}$, RWC of crop residues and soil can be fitted accurately with the same exponential decay function. $R^2 = 0.90$ and root-mean-square error (RMSE) = 0.12, which indicates that $R_{2005}$ had similar simulation effects on the RWC of crop residues and soil. Therefore, the $R_{2005}$ can be used to simulate the RWC of the mixture of crop residue and soil with the same curve. In Figure 8b,c, because the saturation plateau occurred when the RWC was 1, two ratio water indices used piecewise functions to simulate water. When $R_{1.6}/R_{1.5} < 1.54$, RWC = 1.76 * $(R_{1.6}/R_{1.5}) - 1.72$. When $R_{1.6}/R_{1.5} \geq$ 1.54. RWC = 1, $R^2 = 0.75$, RMSE = 0.17; similarly, when $R_{1.6}/R_{2.0} < 2.53$, RWC = 0.59 * $(R_{1.6}/R_{2.0}) - 0.50$. When $R_{1.6}/R_{2.0} \geq 2.53$, RWC = 1. $R^2 = 0.84$, RMSE = 0.14.

### 3.6. Inversion of RWC$_m$

Under actual field conditions, crop residue and soil are mixed together. Therefore, it was necessary to verify the inversion effect of $R_{2005}$ obtained from laboratory experiments on a field with mixed moisture. Figure 9 shows the fitting results of RWC$_m$ and the $R_{2005}$, $R_{1.6}/R_{1.5}$, $R_{1.6}/R_{2.0}$.

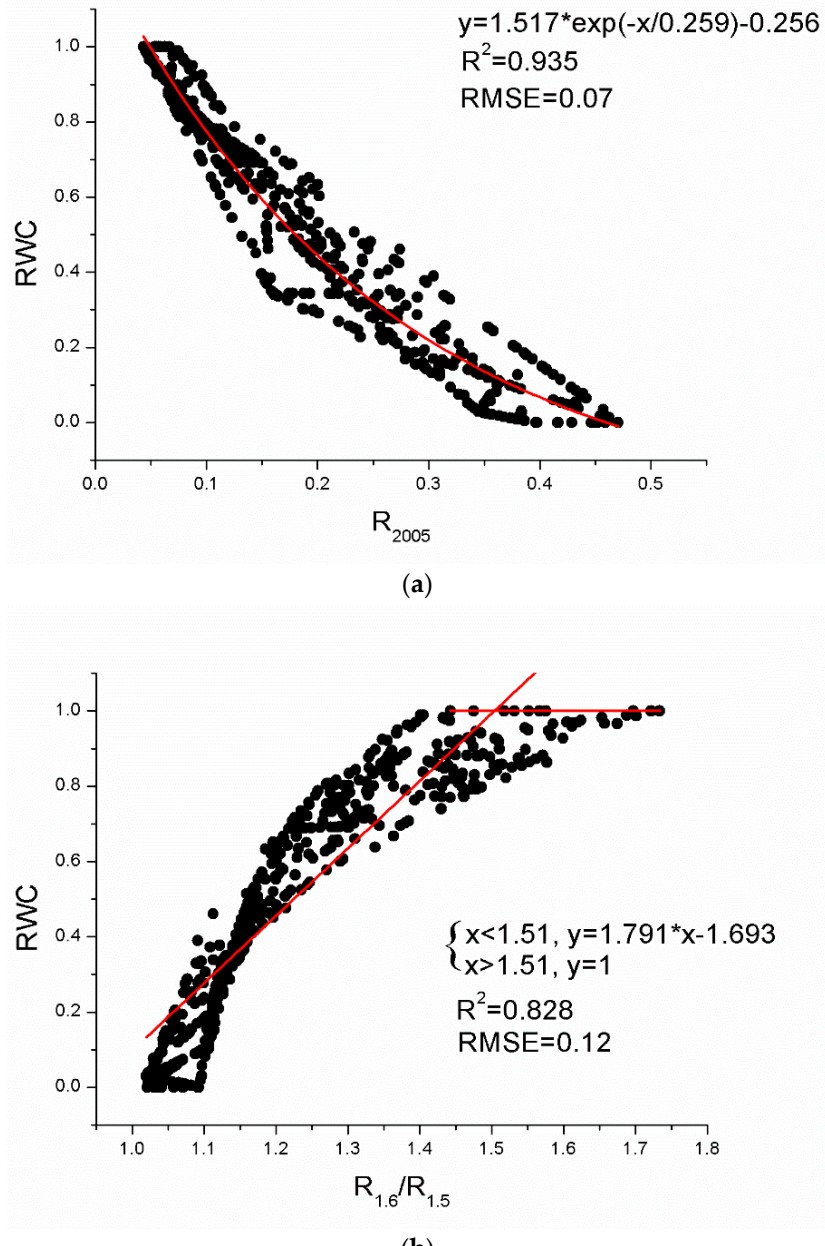

(**a**)

(**b**)

**Figure 9.** *Cont.*

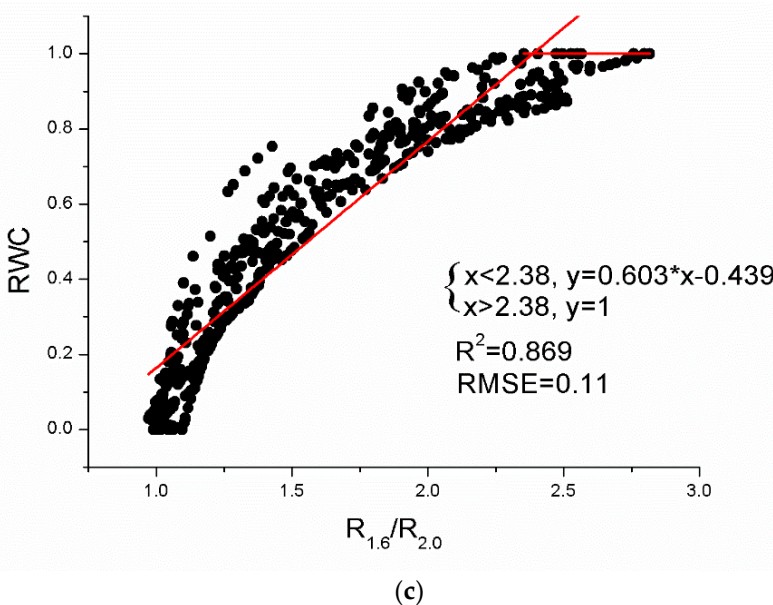

(**c**)

**Figure 9.** $RWC_m$ as a function of the sensitive band and two water indices: (**a**) $R_{2005}$ (**b**) $R_{1.6}/R_{1.5}$ (**c**) $R_{1.6}/R_{2.0}$.

It was obvious that $R_{2005}$ was the best band for inverting $RWC_m$. $R^2$ is 0.935, RMSE is 0.07. Of the two water indices, $R_{1.6}/R_{2.0}$ was more effective than $R_{1.6}/R_{1.5}$ in simulating $RWC_m$. Through comparison, it can be seen that the $RWC_m$ inversion effect is significantly improved after using the $R_{2005}$. $R^2$ increased from 0.828 to 0.935 and RMSE declined from 0.12 to 0.07, which shows that the regression fitting degree between the $R_{2005}$ and the actual $RWC_m$ was high, and the gap between the observed values and the true values was small. The use of the $R_{2005}$ had the advantage of precision in the inversion of $RWC_m$ that is moisture mixture of crop residues and soil.

*3.7. Validation*

The research selected 70 sample points to validate the accuracy of models improved by $R_{2005}$, $R_{1.6}/R_{1.5}$ and $R_{1.6}/R_{2.0}$. The results are shown in Figure 10.

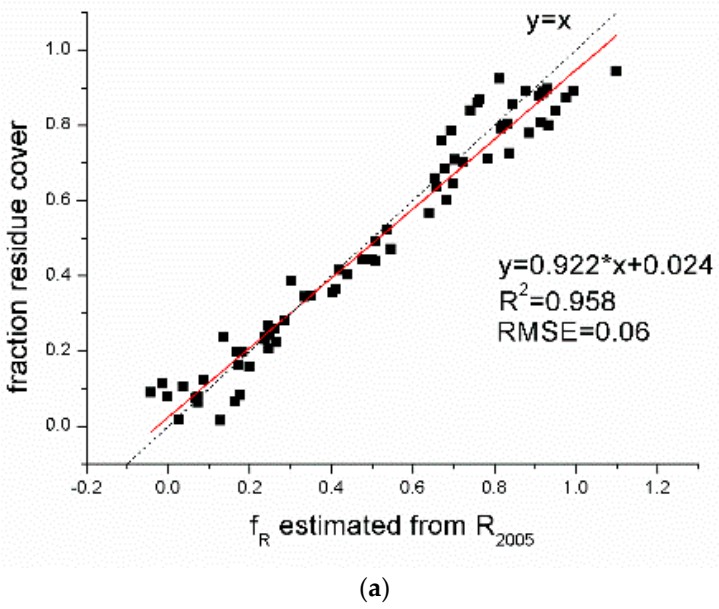

(**a**)

**Figure 10.** *Cont.*

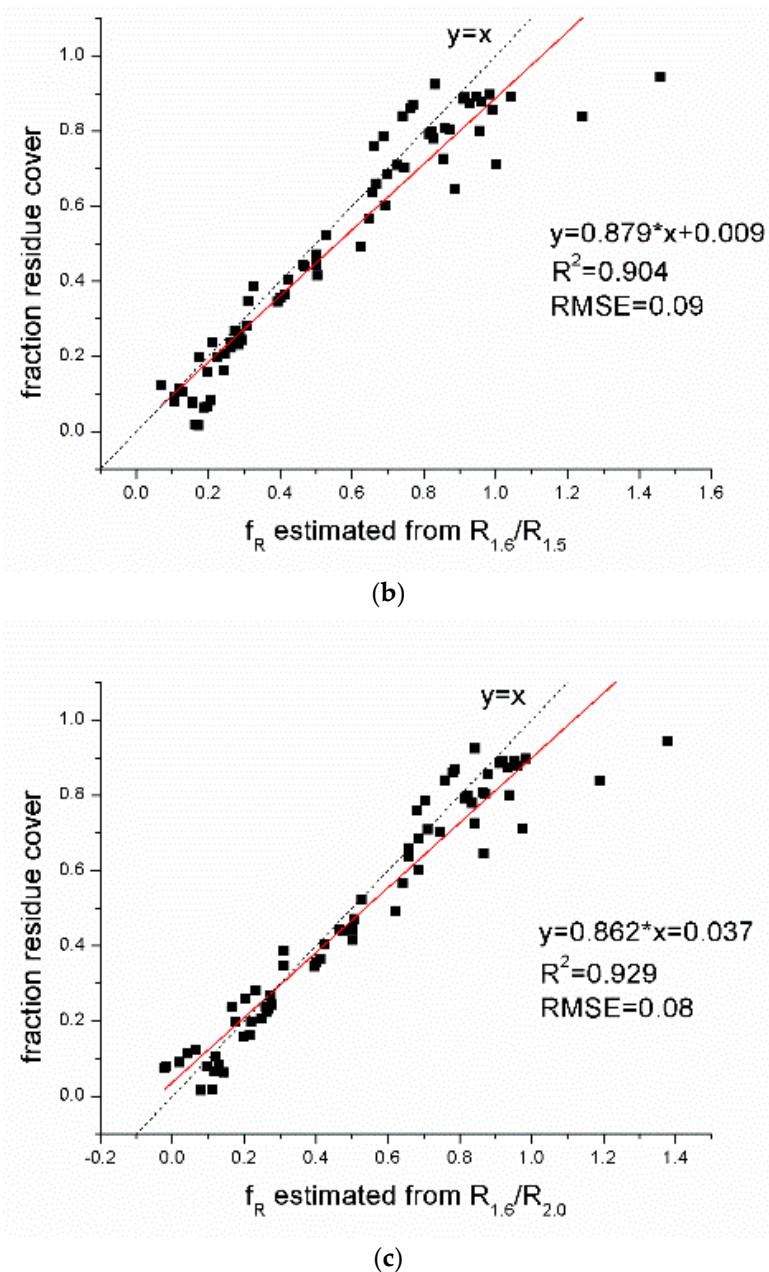

**Figure 10.** The contrast between estimated crop residue cover and true crop residue cover.

The solid lines were linear equations obtained by regression analysis of simulated and true values of crop residue cover. The dotted lines were the $y = x$ standard lines. Compared with the two ratio water indices, the slope of the simulated and true values of crop residue cover using $R_{2005}$ is the closest to 1 and intercept was minimum. $R^2$ was 0.958, and RMSE was only 0.06.

## 4. Discussion

Compared with the two ratio water indices, the $R_{2005}$ had a number of advantages. Unlike the ratio water indices, which required piecewise fitting functions, the $R_{2005}$ only needed a curve to be fitted without impact of saturation plateau. In practical application, $R_{2005}$ was more suitable to be promoted in the field than ratio water indices, due to fewer bands applied and simple and efficient calculation methods. The $R_{2005}$ can quickly and accurately simulate the moisture condition of farmland, especially when the available bands of satellite images or aerial images in the study area are limited.

Moreover, $R_{2005}$ had a higher accuracy than ratio water indices in terms of moisture inversion. $R^2$ increased from 0.828 to 0.935, RMSE declined from 0.12 to 0.07 at the point of $RWC_m$ inversion. At the point of results verification, the highest $R^2$ with 0.958 and the lowest RMSE with 0.06 indicated that the use of the $R_{2005}$ can estimate RWC most accurately and effectively to further improve crop residue cover estimation accuracy. The $R_{2005}$ could minimize the influence of moisture in the estimation process of crop residue cover to a great extent.

Since it is difficult to simulate a realistic varied terrain in a laboratory experiment, the effect of topographic relief was not considered. In fact, studies have shown that topographic fluctuations can affect reflectance [28,29]. When the study area is mountainous, researchers should also consider terrain effect. The experiment samples were collected from Liaocheng City, Huang-Huai-Hai Plain, where the main cultivated soil is the moist soil and the main crop is winter wheat [30–32]. Therefore, only one kind of soil and one kind of crop residue were collected for study. When considering alternative study areas, which are not the Huang-Huai-Hai Plain, the type of soil and crop residues may change. The experiment needs to be redesigned for different study areas. The laboratory experiment was carried out under ideal conditions and there are differences in the realistic fields. The incapability of simulating real field conditions is a limitation of laboratory experiments. In addition, soil salinity, PH value, trace element content, and other variables can also become the future research subjects of crop residue cover factors.

## 5. Conclusions

The influence of water on the estimation of crop residue cover in cultivated land was discussed and eliminated largely. Crop residue cover can be assessed by a linear model with CAI, while moisture would lead to low estimations. A sensitive band, $R_{2005}$, was proposed in this paper to invert the RWC of crop residues and soil and improve estimation of crop residue cover. The $R_{2005}$ was compared with two ratio water indices in many aspects. At the point of validation, the result of $R^2$ was 0.958 and RMSE was only 0.06, proving that the model improved by $R_{2005}$ had the highest reliability. In addition, unlike the piecewise fitting functions of ratio water indices, the $R_{2005}$ only needed a single function for accurate fitting. For the promotion to practical application, the $R_{2005}$ had great advantages, such as simple bands calculation. It could avoid unavailable spectral bands and select appropriate and high-quality data, especially for limited spectral data or spectral channels.

**Author Contributions:** Y.W. was responsible for data processing, primary analysis, visualization and paper writing. S.Z. was responsible for funding acquisition, supervision and paper revision. W.C. had provided great help in the laboratory experiment and data analysis. J.H. had made great contribution in revision and verbal polish. F.P. had help a lot in the laboratory experiment.

**Funding:** This research was funded by the Natural Science Foundation of China (41671429), the National Key R&D Program of China (2016YFB0502503), Natural Science Foundation of Jiangsu province, China (BK20181262) and the ISEF program of KFAS.

**Acknowledgments:** The authors thank for help from Buyun Lei, Huijun Xiao, Yongchao Qu, and Cheng Chen, of School of Geography and Ocean Science, Nanjing University, China for their field measurement work.

**Conflicts of Interest:** The authors declare no conflict of interest.

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
