# Peer review of "A Sensitive Band to Optimize Winter Wheat Crop Residue Cover Estimation by Eliminating Moisture Effect"

_sustainability, doi:10.3390/su11113032_

Round 1

Reviewer 1 Report

Remote sensing residue cover accurately on the ground is based on a spectral absorption feature at 2100-nm wavelength. Soils do not have an absorption feature at this wavelength. However, water does have strong absorption features about 2100-nm wavelength. Monitoring residue cover in the springtime is one of the best methods to assess the amount of conservation tillage. In the springtime, soil and residue moisture content is highly variable, so remote sensing of residue cover must include corrections for moisture content. Experimentally, the study is very similar to previous work, whereas the discovery that reflectance at a wavelength of 2005 nm would provide a better correction is novel.

Major comments:

1. Reflectance at a single wavelength have problems with different backgrounds and conditions, including different soils, because radiance is measured directly, and reflectance is calculated from the incident light level. Variations of soil surface topography will change the incident light level by the cosine of the incidence angle.  From a satellite or airborne perspective, variations of surface topography will result in different reflectances.  Small scale variations in surface topography will be missed in digital elevation data, so soil and vegetation indices are still required to normalize the data. This point is much more important to discuss compared to missing data about a RWC of 50% (L 340-344).  Identifying better wavelengths for moisture correction is worthy of publication, but the algorithms need much more work to be practical.

2. On lines 74 to 78, you state existing indices are not accurate for moisture correction. Is there any information on what the required minimum accuracy is? While the changes in R2 and RMSE are large, do the changes achieve greater accuracy? Accuracy is usually assessed by skill in classification, so this study can’t answer the question.  However, this point is important for discussion.

3. Because of the limited objective, having a limited experimental design is acceptable. However, a thorough study would need multiple replicates (different rates of evaporation), multiple soil types (different brightnesses), and different crops (different lignin and cellulose contents). This is another point for discussion.

Minor comments:

General: there is a big difference in significant digits from 1.6 micrometers compared to 2005 nanometers wavelength. The difference becomes confusing when the units of wavelength are not provided. Changing to the same units won’t be enough, since R2005 becomes R2.005 emphasizing the number of significant digits. Would 2.0 micrometers suffice, or do you really need the accuracy and precision of 1 nanometer?

General: when including a name in a citation (e.g., Zheng et al. on line 44), the citation number to the reference needs to be placed right after the names (e.g., Zheng et al. [12]).

General: MDPI journals do not charge for length, so there is no cost to make each figure larger. Figures 2, 3, 5, 6, 8, and 9 are very small and the details are hard to see.  Figures 1, 4, and 7 have acceptable size. 

General: there is a space between the numerals and the units of measure (e.g. 2005 nm)

Introduction: Figures 2 and 3 show the changes in spectral reflectance for residue and soil, but they don’t show the relationships of these spectra to the various absorption features and spectral bands for the different satellites (Thematic Mapper, OLI, and WorldView-3). The readership of Sustainability is very diverse so more there needs to be more introduction to remote sensing. An introductory figure with typical leaf, soil, and residue spectra and indicators for sensor band wavelengths would really help explain the key points in the paragraph (lines 56-73).

Introduction: Please consider moving Table 2 here and including the other spectral indices mentioned in the text.

Equation 1: “turgid” is only applicable to water inside a semipermeable membrane (i.e. living cells). “max” would work nicely.

Lines 113-121: what was the duration of the experiment?

L 130-160: Italics capital R is used for both reflectance and residue. Please consider using the lower-case italics Greek letter, rho (ρ) for reflectance.

L 172-173: what type of regression methods?

L 175-177: how was sensitivity calculated?

Figures 2 and 3: legends should be consistent (RWC = 1.00 to RWC = 0.00, and round RWC = 0.028 and 0.007 to 0.03 and 0.01, respectively).

L 204-214: humidity is the moisture content of the air, just use moisture or water content here.

Table 1: “slope” in the equation is misspelled.

L 340-344: evaporation rate could have been controlled or sampling intervals could be shorter. I suggest deleting this paragraph.

References: what does the “[J]” after the article title signify? References 4, 6, 10, 12, 15, 17, 18, 19, 21, 22, 23, and 24 have problems with either names, missing information, punctuation, capitalization, or journal names are abbreviated. Please check each actual journal article instead of relying on some database (e.g. Google Scholar).  Watch for automatic formatting problems (e.g. III instead of Iii).

Author Response

Thank you for the valuable comments. Our responses to reviewers’ comments are written in red color.

Major comments:

1. Reflectance at a single wavelength have problems with different backgrounds and conditions, including different soils, because radiance is measured directly, and reflectance is calculated from the incident light level. Variations of soil surface topography will change the incident light level by the cosine of the incidence angle.  From a satellite or airborne perspective, variations of surface topography will result in different reflectances.  Small scale variations in surface topography will be missed in digital elevation data, so soil and vegetation indices are still required to normalize the data. This point is much more important to discuss compared to missing data about a RWC of 50% (L 340-344).  Identifying better wavelengths for moisture correction is worthy of publication, but the algorithms need much more work to be practical.

Reply: Thank you for your valuable comments. The effect of terrain should be considered in an actual situation due to the difficulty in studying the effect of topographic relief in laboratory experiments. In the discussion section, the limitation of laboratory experiment and importance of terrain have been admitted for readers.

2. On lines 74 to 78, you state existing indices are not accurate for moisture correction. Is there any information on what the required minimum accuracy is? While the changes in R2 and RMSE are large, do the changes achieve greater accuracy? Accuracy is usually assessed by skill in classification, so this study can’t answer the question.  However, this point is important for discussion.

Reply: Thank you for this suggestion. There is no information on the minimum accuracy limitation. While the results showed reflectance at 2005 nm can invert moisture better and improve estimation accuracy than existing indices. As a laboratory experiment, a better accuracy is the goal.

3. Because of the limited objective, having a limited experimental design is acceptable. However, a thorough study would need multiple replicates (different rates of evaporation), multiple soil types (different brightnesses), and different crops (different lignin and cellulose contents). This is another point for discussion.

Reply: Thank you for this suggestion. Because the main cultivated soil is moist soil in the Huang-huai-hai Plain, and the main crop is winter wheat (summer corn has a short growing season and winter wheat is the focus of this paper), the samples include one kind of soil and one kind of crop residue. This problem was explained in the discussion section.

Minor comments:

1. General: there is a big difference in significant digits from 1.6 micrometers compared to 2005 nanometers wavelength. The difference becomes confusing when the units of wavelength are not provided. Changing to the same units won’t be enough, since R2005 becomes R2.005 emphasizing the number of significant digits. Would 2.0 micrometers suffice, or do you really need the accuracy and precision of 1 nanometer?.

Reply: Because laboratory spectral data, instead of satellite images, were used in this manuscript, we chose to use these measurements. The researchers would like to improve accuracy as much as possible and selected the band with the highest sensitivity. While the result will be promoted to satellite images, the bands approached 2000 nm are accepted. The bands approached 2000 nm all have high sensitivity.

2. General: when including a name in a citation (e.g., Zheng et al. on line 44), the citation number to the reference needs to be placed right after the names (e.g., Zheng et al. [12]). 

Reply : The number [12-23] has been placed after the names.

3. General: MDPI journals do not charge for length, so there is no cost to make each figure larger. Figures 2, 3, 5, 6, 8, and 9 are very small and the details are hard to see.  Figures 1, 4, and 7 have acceptable size.

Reply : The Figures 2, 3, 5, 6, 8, 9 and 10 have been magnified.

4. General: there is a space between the numerals and the units of measure (e.g. 2005 nm)

Reply : A space has been placed between the numerals and the units of measure in the full text.

5. Introduction: Figures 2 and 3 show the changes in spectral reflectance for residue and soil, but they don’t show the relationships of these spectra to the various absorption features and spectral bands for the different satellites (Thematic Mapper, OLI, and WorldView-3). The readership of Sustainability is very diverse so more there needs to be more introduction to remote sensing. An introductory figure with typical leaf, soil, and residue spectra and indicators for sensor band wavelengths would really help explain the key points in the paragraph (lines 56-73).

Reply : Crop residue indices built by crop residue sensitive bands for different satellites have been added with words in the introduction section.

6. Introduction: Please consider moving Table 2 here and including the other spectral indices mentioned in the text.

Reply : A table describing water indices mentioned has been added in the introduction. Crop residue indices besides CAI are not the focus in the manuscript and have not been listed in detail.

7. Equation 1: “turgid” is only applicable to water inside a semipermeable membrane (i.e. living cells). “max” would work nicely.

Reply : The “turgid” has been replaced by “max”.

8. Lines 113-121: what was the duration of the experiment?

Reply : About 36 hours.

9. L 130-160: Italics capital R is used for both reflectance and residue. Please consider using the lower-case italics Greek letter, rho (ρ) for reflectance.

Reply : Thank you for this suggestion. The R has been replaced by ρ.

10. L 172-173: what type of regression methods?

Reply : A linear regression was applied for intercept and RWC. An ExpDec1 regression was applied for slope and RWC.

11. L 175-177: how was sensitivity calculated?

Reply : The Pearson Correlation Coefficients between RWC and reflectance were calculated as sensitivity indications.

12. Figures 2 and 3: legends should be consistent (RWC = 1.00 to RWC = 0.00, and round RWC = 0.028 and 0.007 to 0.03 and 0.01, respectively).

Reply : Figures 2 and 3 have been modified.

13. L 204-214: humidity is the moisture content of the air, just use moisture or water content here.

Reply : The “humidity” has been replaced by “moisture”.

14. Table 1: “slope” in the equation is misspelled.

Reply : The spelling mistake has been modified.

15. L 340-344: evaporation rate could have been controlled or sampling intervals could be shorter. I suggest deleting this paragraph.

Reply : The part of evaporation rate has been deleted and other discussion contents were added in discussions.

16. References: what does the “[J]” after the article title signify? References 4, 6, 10, 12, 15, 17, 18, 19, 21, 22, 23, and 24 have problems with either names, missing information, punctuation, capitalization, or journal names are abbreviated. Please check each actual journal article instead of relying on some database (e.g. Google Scholar).  Watch for automatic formatting problems (e.g. III instead of Iii).

Reply : The[J]” means the paper was published on a journal. The references have been checked and modified.

Thanks again for your valuable suggestions. 

Reviewer 2 Report

The manuscript report results about laboratory and modelling investigation about the possible use of specific reflectance spectra and indices to estimate the moisture content on agricultural crop residues covers.

GENERAL COMMENTS

The manuscript deal with a current and interesting topic, especially when considered the raising application of remote (satellite) and proximal (drones and other unmanned vehicles) sensing to collect data and applied models and decision support systems to improve efficiency of the agricultural systems. I suggest the author to include into introduction some words (and references) about this current topics to made robust the justification of the study.

Nevertheless I raise some issues about the content of the paper that authors should consider to improve the value of the manuscript and make it suitable for publication.

All along the manuscript (especially in the introduction) many abbreviations are introduced without explaining their meaning. Please describe it shortly. Include a reference for an exhaustive explanation of and a case of use of indices and/or reflectance spectra.

SPECIFIC COMMENTS

L25 ..another advantages. What does it mean? What is the additional advantage? It is not specified.

L33 ..large number of studies.. Report the most relevant of them including the reference

L41 …is critical to agricultural development. Explain the meaning of this statement and when possible include references

L79 Material and Methods. There are a lack of references supporting the methodology adopted for the study. Please report previous studies where similar methodology have been adopted.

L160 unitary regression analysis+L173 regression methods. Please report previous studies where similar analysis have been adopted

L169-170 Correct “But…slope and intercept OF THE FUNCTION OF THE MODEL” or similar clarification

LL184 …in 2016. Replace with the reference

L185 Validation. On my opinion too many estimation and calculation of parameters for the validation process. Explain better

L230-232 ..different coverages were not collected in the field…actual spectra values in the field were simulated by linear combination. This raise concerns about the possible real field application. It should include as limitation of the study (the laboratory investigation) in the conclusion

L343-344 In addition….cover factor. I cannot understand the meaning of the sentence. Is it a future work statement?

L345 Conclusion. Consider limitation of the study of the laboratory investigation compared to the requirement of effective indices and reflectance spectra required for in field data acquisition for real smart farming technology applications.

The English language has to be revised by a professional.

Author Response

Thank you for the valuable comments. Our responses to reviewers’ comments are written in red color.

General comments:

1. The manuscript deal with a current and interesting topic, especially when considered the raising application of remote (satellite) and proximal (drones and other unmanned vehicles) sensing to collect data and applied models and decision support systems to improve efficiency of the agricultural systems. I suggest the author to include into introduction some words (and references) about this current topics to made robust the justification of the study.

Reply: Thank you for this suggestion. Crop residue models are mostly based on crop residue indices and some new crop residue indices and model are added in the introduction.

2. Nevertheless I raise some issues about the content of the paper that authors should consider to improve the value of the manuscript and make it suitable for publication.

Reply: Thank you for this suggestion. The manuscript has been revised carefully by authors.

3. All along the manuscript (especially in the introduction) many abbreviations are introduced without explaining their meaning. Please describe it shortly. Include a reference for an exhaustive explanation of and a case of use of indices and/or reflectance spectra.

Reply: Thank you for this suggestion. The abbreviations in this manuscript are mostly crop residue indices. The full name has been added in the parentheses after abbreviations. A table describing water indices has been added in the introduction. Crop residue indices besides CAI are not the focus in the manuscript and have not been listed in detail. 

Specific comments:

1. L25 ..another advantages. What does it mean? What is the additional advantage? It is not specified.

Reply: “another advantage” means with a simple band calculation, R2005 is suitable to promote to actual production. In order to make it easy to understand, L25 has been modified to “Another advantage of R2005 is more suitable to promote to actual production because of simple and efficient band calculation.”

2. L33 ..large number of studies.. Report the most relevant of them including the reference

Reply: The references have been supplemented.

3. L41 …is critical to agricultural development. Explain the meaning of this statement and when possible include references

Reply: This sentence is a summary for above content to make the structure complete.

4. L79 Material and Methods. There are a lack of references supporting the methodology adopted for the study. Please report previous studies where similar methodology have been adopted.

Reply: The two papers which adopted laboratory experiment are listed as following:

[22] Miguel Q, Craig D. Spectral Indices to Improve Crop Residue Cover Estimation under Varying Moisture Conditions[J]. Remote Sensing, 2016, 8(8):660-679.

[26] Daughtry C S T, Hunt E R. Mitigating the effects of soil and residue water contents on remotely sensed estimates of crop residue cover[J]. Remote Sensing of Environment, 2008, 112(4):1647-1657.

5. L160 unitary regression analysis+L173 regression methods. Please report previous studies where similar analysis have been adopted

Reply : A linear regression was applied for CAI and crop residue cover. A linear regression was applied for intercept and RWC. An ExpDec1 regression was applied for slope and RWC. Some previous papers are listed:

[22] Miguel Q, Craig D. Spectral Indices to Improve Crop Residue Cover Estimation under Varying Moisture Conditions[J]. Remote Sensing, 2016, 8(8):660-679.

[25] Daughtry C S T, Serbin G, Reeves III J B, et al. Spectral Reflectance of Wheat Residue during Decomposition and Remotely Sensed Estimates of Residue Cover[J]. Remote Sensing, 2010, 2(2):416-431.

[26] Daughtry C S T, Hunt E R. Mitigating the effects of soil and residue water contents on remotely sensed estimates of crop residue cover[J]. Remote Sensing of Environment, 2008, 112(4):1647-1657.

6. L169-170 Correct “But…slope and intercept OF THE FUNCTION OF THE MODEL” or similar clarification

Reply : L169-170 has been modified as required.

7. L184 …in 2016. Replace with the reference

Reply : The reference has been supplemented after “2016”

8. L185 Validation. On my opinion too many estimation and calculation of parameters for the validation process. Explain better

Reply : The main process is:

First, use the sensitive band, R2005, to invert RWCm;

Second, use RWCm to calculate slope and intercept of crop residue cover model;

Finally, use crop residue cover model to estimate crop residue cover.

To make it easy to understand, the validation has been modified.

9. L230-232 ..different coverages were not collected in the field…actual spectra values in the field were simulated by linear combination. This raise concerns about the possible real field application. It should include as limitation of the study (the laboratory investigation) in the conclusion

Reply: The limitation of laboratory experiments has been admitted in discussions.

10. L343-344 In addition….cover factor. I cannot understand the meaning of the sentence. Is it a future work statement?

Reply: Yes, it is a future work outlook. L343-344 has been modified to “In addition, soil salinity, PH value, trace element content and other variables can also become the future research subjects of crop residue cover factors.”

11. L345 Conclusion. Consider limitation of the study of the laboratory investigation compared to the requirement of effective indices and reflectance spectra required for in field data acquisition for real smart farming technology applications.

Reply: The limitation of laboratory experiments has been admitted in discussions.

12. The English language has to be revised by a professional.

Reply: The full text has been revised by an English professional.

Thanks again for your valuable suggestions. 

Round 2

Reviewer 2 Report

The authors adressedd all the comments and questions raised

The manuscript can be published